# The Effect of Chemical Corrosion on Mechanics and Failure Behaviour of Limestone Containing a Single Kinked Fissure

**DOI:** 10.3390/s21165641

**Published:** 2021-08-21

**Authors:** Yulin Wu, Qianqian Dong, Jian He

**Affiliations:** College of Aerospace and Civil Engineering, Harbin Engineering University, Harbin 150001, China; wuyulin@hrbeu.edu.cn (Y.W.); hejian@hrbeu.edu.cn (J.H.)

**Keywords:** kinked fissure, chemical corrosion, failure behavior, coalescence, brittle limestone

## Abstract

This study concerns the influence of chemical corrosion and geometric parameters on the macroscopic damage characteristics of brittle limestone containing a kinked fissure under uniaxial compression. The specimens are prepared in chemical solutions with different NaCl concentrations and pH values. The acoustic emission (AE) technique is adopted to detect the inner distortion of the failure behaviour. The physical process of the crack coalescence of kinked fissures is synchronously captured by a high-speed camera. Seven failure patterns are identified based on the final failure mode and the failure process. Furthermore, the stress intensity factor of kinked cracks under chemical corrosion is obtained by a theoretical analysis. Chemical erosion with an acidic solution has a relatively strong effect on the compressive strength of the tested specimen, while the initial crack angle is not affected by short-term chemical corrosion.

## 1. Introduction

Natural rocks contain a large number of cracks and joints with irregular geometries, which are usually exposed to complex environments such as groundwater, rivers and oceans. Long-term contact between rock surfaces and different aqueous solutions causes the rock surface to become susceptible to chemical corrosion. These irregular cracks and chemical corrosion essentially determine the coalescence behaviour and mechanical behaviour of the rock, which are of great significance for assessing safety in rock engineering. Moreover, rocks are easy to compress in human engineering activities. In the process of compression, the connection and propagation of cracks have an important impact on the quality of engineering and geological environment. Therefore, it is of great practical significance to study the effect of chemical corrosion on the mechanical behaviour of rocks with pre-existing fissures under compression. In the past several decades, related research has mainly focused on two aspects, i.e., the influence of straight cracks on the coalescence behaviour of rock and the effect of chemical corrosion on the mechanical behaviour of rocks. Extensive experimental investigations [1,2,3,4,5,6,7,8,9] and numerical simulation analyses [10,11,12,13,14,15] have been conducted for fractured rock under compression or intact rock after chemical corrosion. However, few experimental studies considering the effects of the above two aspect factors on rocks have been performed.

It is worth noting that rock masses do not always have straight fissures and, generally, have other forms of cracks, such as curves, crossings, or branch cracks. Kinked cracks are common cracks in nature and have attracted the attention of many scholars. Vitek [16] developed a numerical method for calculating the stress intensity factors of different kinked cracks. Liu et al. [17] investigated the strength properties of rock masses containing intermittent cross joints with rock-like materials. Isida et al. [18] utilized a numerical method to solve the basic problems of an infinite plate with existing branch cracks. Chaker and Barquins [19] carried out a number of uniaxial compressive tests on a PMMA plate with pre-existing defects to investigate the sliding effect on branch cracks. Li et al. [20] performed uniaxial compressive tests on marble samples with pre-existing zigzag-type cracks. It is worth nothing that AE technology was used during this research. Meggiolaro et al. [21] used a special finite element method to investigate the tensile load path and the corresponding stress intensity factor on kinked cracks and branch cracks under tensile loading. Zhang et al. [22] further studied the crack coalescence mechanism of X-type fissures by changing the primary and secondary crack inclinations.

Apart from investigating the influence of pre-existing fissures on the failure mode, the influence of chemical solutions on the mechanical behaviour of rock is of great significance. Atkinson et al. [23] studied the effects of deionized water, HCl and NaOH on the crack growth rate, stress intensity factor and stress intensity coefficient of quartz and found that the chemical composition has a controlling effect on crack propagation. Karfakis et al. [24] studied the effect of a chemical solution on the fracture toughness. Based on the experimental results, a possible intrinsic mechanism for the modification of rock properties in chemical solutions was formulated. Tang and Wang [25] summarized the mechanical effect of the chemical action of water-rock interactions on the deformation and failure of rocks. Tang et al. [26] conducted uniaxial compressive strength tests on granite, red sandstone and limestone under different chemical properties of aqueous solutions with different circulating flow rates. Based on the test results, it was found that the strength of rock decreases obviously after hydrochemistry and that water- rock chemistry has a strong time effect on rock mechanics. Li et al. [27] proposed a chemical damage model to describe the process of corrosion of sandstone and the degradation of the rock deformability and strength by performing acidic corrosion tests on sandstone samples. Feng et al. [28,29] carried out a series of experimental studies to investigate the effects of chemical erosion on the mechanical behaviour of sandstone, limestone and granite. The experimental results indicated that the deterioration of rock mechanics properties depends on chemical ions, ion concentrations, pH values and mineral components of rock. Ding and Feng [30] further replaced the test object with limestone samples containing two straight fissures or three straight fissures. The chemical solution was also replaced by NaCl. According to the experimental results, the rock bridge lap mode with multiple fissure rocks was discussed, and the fracture criterion of rocks with multi-pre-existing fissures under chemical corrosion was proposed. Yao and Feng [31] carried out a meso-mechanical experimental study on limestone with a single straight fissure under coupled chemical corrosion (Na_2_SO_4_) and water pressure. The results revealed that the structure and composition of rocks change after chemical erosion, and this change results in an increase in heterogeneity and a decrease in the strength and modulus of elasticity. Han et al. [32] conducted a conventional triaxial compression test on sandstone after immersion in different chemical solutions (Na_2_SO_4_ and Na_4_HCO_3_) and found that sandstone samples tend to transform from brittle to ductile after chemical corrosion. However, the main contributions in most of the above experimental studies simultaneously considering the influence of fissures and chemical corrosion on the rock are mainly limited to specimens containing straight fissures, whereas research on the mechanical behaviours and the coalescence behaviours of natural rock samples with kinked fissures under uniaxial compression remains scarce.

Therefore, the mechanical properties and macroscopic damage of limestone under chemical corrosion are studied based on pre-existing kinked fissures. Moreover, an acoustic emission technique is used to obtain the real-time crack coalescence process and internal damage for limestone containing a single kinked fissure. The emphasis of the present research is helpful for predicting the kinked crack coalescence behaviour under the influence of a chemical environment.

## 2. Experimental Set-Ups

### 2.1. Specimen Preparation

The samples were selected from limestone, containing 68% calcite, 30% biochip, 1% quartz and 1% opaque metallic mineral. The specific content of each mineral and basic physical properties are listed in Table 1. The limestone material is sensitive to acidic liquid, so this sample is used to study the mechanical damage and crack coalescence behaviour in different chemical erosion environments. The limestone, with a microcry talline and blocky structure, is a fine-grained heterogeneous material with an average density of approximately 2340 kg/m^3^.

In this study, the size of the limestone samples is rectangular with a height of 100 mm, width of 50 mm and thickness of 15 mm, which is in accordance with the height to width ratio of the tested sample suggested by the International Society for Rock Mechanics [33]. The kinked fissures of the specimen are made with a high-pressure water jet cutting machine. The width of the fissure is approximately 1.5 mm. Figure 1 shows the expected specimens and the actual specimens. The kinked fissure consists of one main fissure and two branch fissures. The angle between the branch fissure and the main fissure extension is defined as *α*. To investigate the effect of the prefabricated kinked fissure geometry on the strength and failure pattern of limestone under uniaxial compression, the value of *α* is the only variable of the geometric characteristics of the kinked fissure. Detailed sizes of the kinked fissures and specific shapes of the samples with different geometries are provided in Figure 2.

### 2.2. Preparation of the Chemical Solutions

In this experiment, three different solutions with pH values of 2, 7 and 12 were prepared to study the effects of chemical erosion. The concentration of NaCl solution was designed to be 0.1 mol/L and 1 mol/L, and three groups of specimens were respectively merged into different NaCl solution. The strength and failure behaviour of limestone samples under uniaxial compression were studied by investigating the effects of three variables, namely fissure shapes, solution concentration and pH values. The specific chemical configuration is listed in Table 2.

Before the test, the comparison specimens were removed from the same solution, washed with water and then put into an oven with the temperature controlled at 105 °C to dry for 24 h. After cooling, all specimens were transferred to a sealed container for vacuuming for approximately 10 min and then immersed in the different chemical solutions in listed in Table 3, for 45 days. It should be noted that three parallel tests were repeated for each group.

### 2.3. Testing Procedure

In the present experimental study, uniaxial compression tests were conducted on intact limestone and limestone samples with prefabricated fissures. The specimens were loaded uniaxially in a mechanical servo-controlled testing system with a maximum loading capacity of 1000 kN, which could record the load-displacement information and plot curves in real-time while conducting uniaxial compression experiments, as is shown in Figure 3. The back of the specimen is closely connected with a fixation device using glue. The lower surface of the sensor is connected to the fixing device with a coupling agent. The upper surface is fixed with bolts and connected to acoustic emission through a signal amplifier. To investigate the crack propagation characteristics and the fracture mechanism, limestone samples containing kinked fissures were monitored by acoustic emission technology during partial uniaxial compression tests. From Figure 3, the acoustic emission (AE) counts were recorded by a DS5-32B full-information AE measuring system. The frequency of the AE system can reach 3 MHz. When the acoustic signal was captured. By the AE measuring system, the high-speed camera would monitor the font face of specimens simultaneously. The high speed camera has a resolution of 4 million and a maximum shooting speed of 80,000 frames per second. Once the coalescence crack(s) developed or failure of samples occurred, images would be captured by triggering the camera manually. During the uniaxial compression tests, the loading rate was fixed at 0.2 mm/min in a way of displacement control.

## 3. Experimental Results

### 3.1. Strength Characteristics after Chemical Corrosion

The peak stresses of the six types of kinked fissures after exposure to different chemical corrosion environments are shown in Table 3. It should be noted that *ρ* indicates the degree of damage to the peak strength of the rock after chemical corrosion. *σ* represents the peak stress of the specimen in a natural environment, and *σ*′ represents the peak stress of the specimen after exposure to different chemical corrosion environments. To comprehensively consider the influence of the branch fissure angles, pH values and ion concentration on the peak stress, Figure 4 was plotted based on the data in Table 3. The influence of various variables on the rock peak stress will be analysed in detail as follows.

Table 3 shows that the values of *ρ* are similar under the same chemical environment. Therefore, the change in the values of *ρ* is used to estimate the influence of the pH values and ion concentration on the peak stress. First, taking the case of 1 mol/L of NaCl as an example, the effect of the pH value of the solution on the experimental results is discussed. By comparing the range of *ρ* for different pH values, it can be found that the neutral solution (1 mol/L NaCl, pH = 7) has the least corrosion effect on the peak strength of the limestone, and the *ρ* values range from 6.11 to 8.84. The alkali solution (1 mol/L NaCl, pH = 12) has medium corrosion, and the *ρ* values range from 13.05 to 16.13. Last, acid solution (1 mol/L NaCl, pH = 2) has the greatest corrosion effect on the limestone, and the *ρ* values range from 20.52 to 26.57.

In addition to the pH value, the ionic concentration of the solution as the second variable is also noteworthy. In the case where the pH value and the shape of the specimens are the same, the peak stress of the rock decreases slightly with increasing ion concentration by comparing the *ρ* values. For example, in the 0.1 mol/L NaCl, pH = 7 environment, *ρ* is 4.53~7.91. In the 1 mol/L NaCl, pH = 7 environment, *ρ* is 6.11~8.84. Compared with the value of *ρ* in distilled water (4~5.86), the increase in the value of *ρ* is small; that is, the ion concentration has little influence on the peak strength of the rock in this experiment.

In addition to the influence of the abovementioned chemical factors on the peak strength of rocks, the branch fissure angle is also one of the important factors that affects the peak strength of rocks. From Figure 4, it is easy to conclude that the peak stress generally decreases as the angle *α* decreases. Moreover, when *α* experiences clockwise rotation (“+”), the peak strength is much greater than when *α* experiences anti-clockwise rotation (“−”), which may result from the change in the stress concentration position causing a change in the crack initiation position. Eventually, the cracking stress and peak stress of the samples are affected. Macroscopically, the crack and void interfaces between the fracture surface and aqueous solution increase and expand, as shown in Figure 5. It can be clearly seen from Figure 5 that in the natural environment, the fracture surface is relatively flat, and the small pores are sparsely distributed on the fracture surface. After the samples are exposed to a neutral environment (distilled water, 0.1 mol/L NaCl pH = 7 and 1 mol/L NaCl pH = 7), the pores expand in volume, and the number of pores gradually increases. After the limestone samples are subjected to alkaline or acid corrosion (1 mol/L pH = 12 NaCl and 1 mol/L pH = 2 NaCl), with a significant increase in the pore size, the fracture surface also becomes rougher. Thus, the size and quantity of the pores increase with further action of chemical corrosion, and the fracture interfaces in contact with the chemical solutions also become rougher.

Therefore, based on chemical tests, the chemical environment has a minor effect on the compressive strength of rock. The acid environment makes the specimen more prone to failure and the order of the influence of chemical corrosion on the failure strength of limestone can be regarded as acidic environment > alkaline environment > distilled water > natural environment.

### 3.2. Acoustic Emission Behaviours of Kinked Fissures

As is shown in Figure 6 and Figure 7, according to the characteristics of the stress-time curves for the intact sample and pre-existing fissure samples with varying *α* values, the axial stress-axial strain behaviour containing a single kinked fissure can be approximately divided into five stages, i.e.,; the primary microcrack compaction stage (o–a), elastic modulus stage (a–b), stable crack growth stage (b–c), unstable crack growth stage (c–d) and global post-peak failure stage (d–e) have different AE features in their respective stages. The corresponding AE behaviour is also presented in Figure 6 and Figure 7.

At the primary microcrack compaction stage, the specimen is less stressed and the strain changes greatly, which results from the primary fissure and pores being compacted at this stage. Therefore, there are only a few or no acoustic emission events at this stage, and the value of the acoustic emission count is also smaller than that in the active period.

At the elastic modulus stage, AE counts are also maintained at a low level. Figure 6 and Figure 7 show the above-mentioned AE behaviour of the intact samples and the flawed limestone samples containing a single kinked fissure in the elastic stage. The stress-time curve and acoustic emission curve all follow a similar pattern despite the angle change. Strain energy accumulates in this stage and will be released in the next stages. At the stable crack growth stage, new cracks inside the limestone initiate, and the primary cracks begin to expand, which causes the stress-time curve to gradually lose linearity. Point b usually corresponds to an obvious AE event, which means that AE technology can be used to determine the crack initiation stress under uniaxial compression. After point b, AE events begin to increase with the crack initiation and growth. The values of the AE counts also increase gradually during this stage. At the unstable crack growth stage, the stress-time curve in this stage rises with a local fluctuation or sudden drop. The maximum AE counts often occur at this stage accompanied by a sudden drop in the curve. It seems that the AE events are denser than those during the previous stages, and the AE counts are still maintained at a high level. As is shown in Figure 7, the AE counts of the sample with prefabricated cracks is significantly larger than that of the intact limestone sample. In the stage o-a, the AE counts of (a)~(f) remain low. When is −30°, 30°, 45°, 60°, at the stage of a-b, an obvious acoustic emission signal appears at point b, and the AE counts becomes more intensive. Considering the complex failure pattern of the sample, it indicates that certain micro-cracks also appear in the elastic stage. At the stage of b–c, when is −60°, 30°, 45°, 60°, the peak value of AE counts appears near point c, indicating that there were many micro-cracks in the sample at this time, which didn’t appear together with the fracture stress represented by point d. The reasons for this phenomenon are complicated, which may be related to the material of the limestone sample. The non-compactness of the material itself has a certain influence on the failure behaviour of the rock. The stage c-d is the unstable crack growth stage, at which the stress-time curve has a certain fluctuation, and the stress in (d), (e), (f) decreases first and then increases, which is obviously different from the curve of the intact sample. During this period, AE counts is relatively dense and increases to a certain extent. Finally, the samples fail in the d-e stage, and the stress and AE counts decreases rapidly.

To further study the relationship between the failure process of the limestone sample and the stress-strain curve under uniaxial compression, taking *α* = 45° and *α* = −45° in the natural environment as examples, the whole rock failure process was recorded by a high-speed camera. The relationship between the typical stress-strain curve and the crack propagation process when *α* = 45° is illustrated in detail in Figure 8. As the strain increases, the stress-strain curve shows a linear growth trend after the primary crack compaction phase. As is shown in Figure 8, with the stress increasing to point b (22.9 MPa), crack pattern 1 initiates at the upper and lower tips of the branch fissure. As mentioned above, the stress-strain curve does not fluctuate significantly, but point b corresponds to a distinct AE signal. On account of cracks 1*^a^* and 1*^b^* being too thin, naked eyes can’t capture them. When the load reaches 35.7 MPa, apart from the length of cracks 1*^a^* and 1*^b^* increasing, there is new crack 2 appearing which initiates from the top of the crack tip. With the further increase of the axial deformation, the stress rises to point c (45.85 MPa). At this moment, the stress drops from 45.85 MPa to 44.77 MPa, due to far-field crack 3 generating from the upper surface of the specimen. Then, the sample still maintains a certain bearing capacity after the axial stress increases to the peak point d (52.26 MPa). With the formation of crack 4 from the lower tip of the branch fissure, the corresponding axial decreases from 52.26 MPa to 44.47 MPa. As the axial strain continues to increase, the stress reaches 48.45 MPa. At this time, crack coalescence behavior is observed, and there is a rapid decline of the stress from 48.45 MPa to 45.48 MPa. Afterwards, as the increase of the strain continues, accompanied by a loud sound, the sample loses its bearing capacity.

Figure 9 also presents the whole crack coalescence process of the limestone sample when *α* is −45°. Note that when the sample is loaded to point b (17.37 MPa), cracks 1*^a^* and 1*^b^* initiate from the junction of the branch fissure and the main fissure, which is different from the previous example. At this time, the stress-strain curve fluctuates slightly. After point b, with increasing axial strain, cracks 1*^a^* and 1*^b^* develop along the axial stress direction. When the stress increases to 27.56 MPa, crack 2 emanates from the lower tip of the branch fissure and propagates towards the upper edge of the sample. The slope of the corresponding stress-strain curve decreases, and the specimen tends to change from elastic to plastic. Afterwards, when the sample is loaded to point c (33.65 MPa), crack 3 is produced from the upper tip of the branch fissure and propagates towards the upper edge of the sample. The corresponding stress also drops rapidly from 33.65 MPa to 28.21 MPa. After point c, before point d, the sample exhibits greater plasticity as the axial strain increases. Far-field crack 4 also appears on the surface of the specimen during this period. When the axial stress is loaded to peak point d (37.45 MPa), crack coalescence behaviour is observed at this time, and more secondary cracks also occur during period d~e, which causes the stress to drop rapidly from 37.45 MPa to 20.69 MPa. Subsequently, the stress-strain curve exhibits a brief rise and then rapid decline. Then, the sample loses its load carrying capacity.

### 3.3. Acoustic Emission Behaviours of Kinked Fissures

Based on the crack initiation position and propagation mechanism of all limestone specimens under chemical corrosion, seven different crack types are summarized in Figure 10 [34,35]. The detailed characteristics of each crack type are described below:

Crack pattern I: The upper and lower tips of branch fissures are the points from which tensile cracks emanate. For wing cracks, their propagation usually begins along the vertical direction to the main fissure and then gradually develops towards the axial stress direction.

Crack pattern II: Tensile cracks are generated from the tips of branch fissures while the wing cracks propagate just opposite from crack pattern I.

Crack pattern III: The junction is where the branch fissure and the main fissure meet (hereafter called the junction), from which tensile cracks initiate from. The direction of the wing crack in the initial condition is approximately vertical from the main fissure. As the load increases, the cracks develop and gradually propagate to the axial stress direction.

Crack pattern IV: Tensile cracks emanate from the junction, whereas the propagation of the wing cracks is in the opposite direction of crack pattern III.

Crack pattern V: The junction and the main fissure are the main places where lateral cracks initiate. The crack expansion direction of crack pattern V is approximately horizontal to the lateral surface of the specimen.

Crack pattern VI: For shear cracks, the junction is the main location for crack generation and the tips of the branch fissures are another location where shear cracks emanate. The cracks coalesce parallel to the direction of the main fissure under the increasing load.

Crack pattern VII: Far-field cracks, of which the location generally keeps a certain distance from the kinked fissure, are mainly vertical when the load increases, while a small number of them are horizontal.

According to the results of this experiment, chemical factors also have a minor impact on the failure mode of limestone specimens with a prefabricated single kinked fissure. As is shown in Figure 11, the limestone samples exposed to different chemical corrosion environments present a variety of typical final failure modes. All the failure modes of the limestone samples in different chemical environments are summarized in Table 4 [36].

As the degree of chemical corrosion increases, the failure modes of the limestone samples are more abundant. Taking *α* = +45° as an example, we found the influence of the chemical factors on the failure behaviour of rock with prefabricated fissures. In natural environments and in neutral aqueous solutions (distilled water, 0.1 mol/L NaCl, pH = 7 and 1 mol/L NaCl, pH = 7 solution), there are a total of three types of crack patterns. However, in acidic and alkaline aqueous solutions (1 mol/L NaCl, pH = 12 and 1 mol/L NaCl, pH = 2 solution), the total failure mode can reach four types. In addition, compared with the ion concentration, the pH value has a greater influence on the failure mode. According to Table 4, crack pattern 4 only occurred in the 1 mol/L NaCl, pH = 2 situation, which means that long-term chemical corrosion may cause a new crack pattern. 

With the increase in chemical corrosion, more secondary cracks appear in the single limestone sample, as shown in Figure 11. The above phenomenon may be caused by the following two reasons. On the one hand, long-term chemical corrosion aggravates the overall heterogeneity of rock materials, leading to more crack patterns appearing in limestone samples, as shown in Figure 11. On the other hand, the kinked fissure surface contacts the chemical solution for a long time, resulting in chemical reactions that expand the pores and increase the porosity and unevenness, as shown in Figure 5.

In addition, the limestone samples tend to transform from brittle to ductile materials after chemical corrosion. Taking *α* = 45° as an example, the initial cracks developed rapidly, but little or no secondary cracks occurred under natural conditions. Shortly after crack coalescence behaviour occurs in the limestone sample, the sample immediately loses its bearing capacity accompanied by a distinct sound. However, more secondary cracks may appear in the limestone samples after chemical corrosion. At the same time, as secondary cracks develop, the specimens also continue to have a strong bearing capacity.

## 4. Conclusions

To study the mechanical damage and crack coalescence behavior of limestone containing a pre-existing single kinked fissure, with the aid of the AE technique, the limestone samples exposed to different chemical environments under uniaxial compression are observed and analyzed. The results can be generally concluded as follows: 

1. The inclination angle of a branch fissure has an important influence on the deformation and strength of rock, and the peak stress decreases as the angle α decreases. Moreover, the chemical environment has a minor effect on the compressive strength of rock based on chemical tests. The acid environment makes the specimen more prone to failure, and the order of the influence of chemical corrosion on the failure strength of limestone can be regarded as acidic environment > alkaline environment > distilled water > natural environment.

2. The AE characteristics can be divided into three periods, i.e., the quiet period, active period and remission period, which correspond to the primary microcrack compaction stage and the elastic modulus stage, the stable crack growth stage and the unstable crack growth stage, and the global post-peak failure stage, respectively. Moreover, AE technology can be used to determine the crack initiation strength and combined stress curve of the samples under uniaxial compression. If the AE counts are dense and maintain a high level under uniaxial compression, the specimens are usually close to failure. 

3. Seven failure patterns were identified and can consist of the final failure mode and describe the failure process of samples with a single kinked fissure. The non-tip- cracking pattern is mainly responsible for the coalescence, while the main fissure angle is kept constant but has a variable branch fissure angle (α = −45°, −60°).

## Figures and Tables

**Figure 1 sensors-21-05641-f001:**
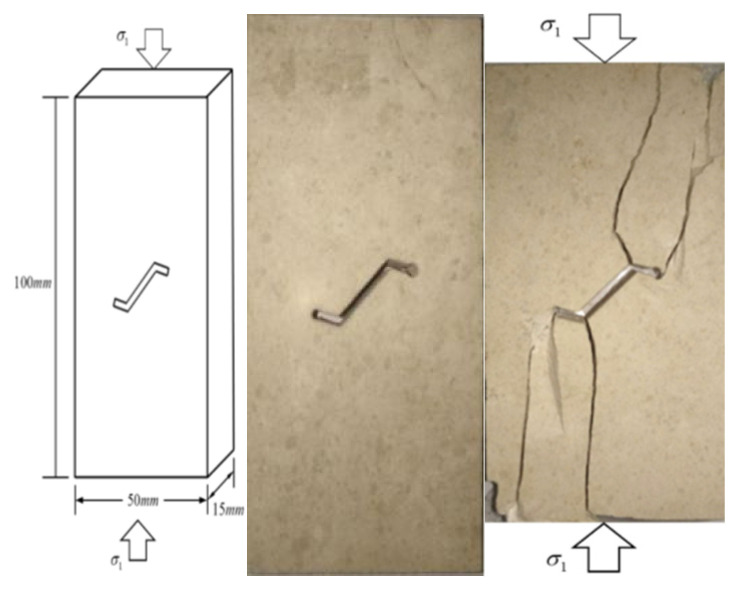
The dimension of limestone specimen containing single kinked fissure under uniaxial compression, actual specimen containing single kinked fissure machined by high pressure waterjet cutting and final failure mode of actual specimen.

**Figure 2 sensors-21-05641-f002:**
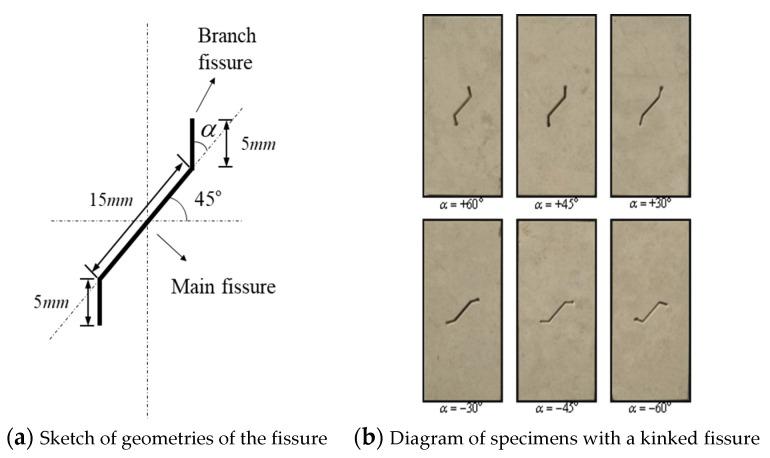
Geometric characteristics of single kinked fissure in the limestone samples, in which *α* is angle between branch fissure and extension line of main fissure.

**Figure 3 sensors-21-05641-f003:**
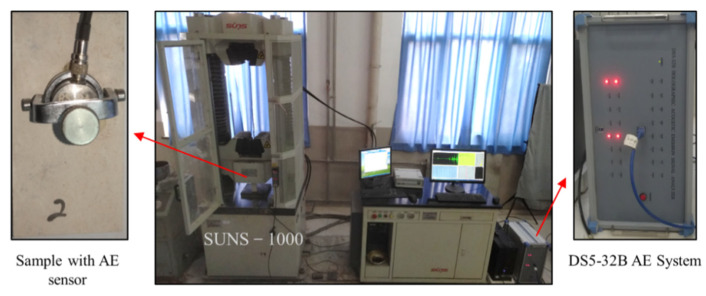
SUNS-1000 mechanics servo-controlled testing system and DS5-32B full-information AE measuring system.

**Figure 4 sensors-21-05641-f004:**
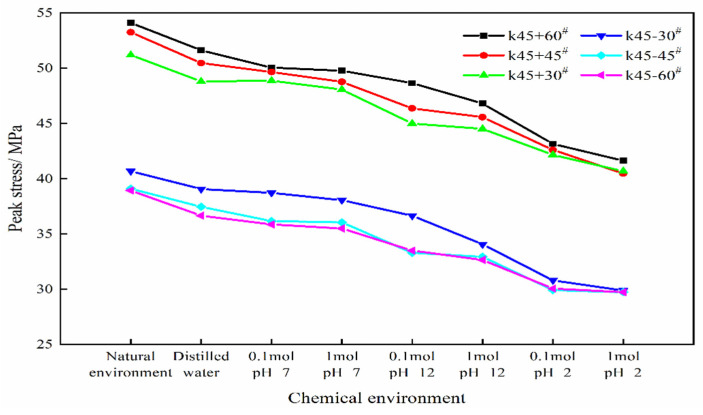
Peak Stress (strength) of limestone specimens containing single kinked fissure after different chemical corrosion.

**Figure 5 sensors-21-05641-f005:**
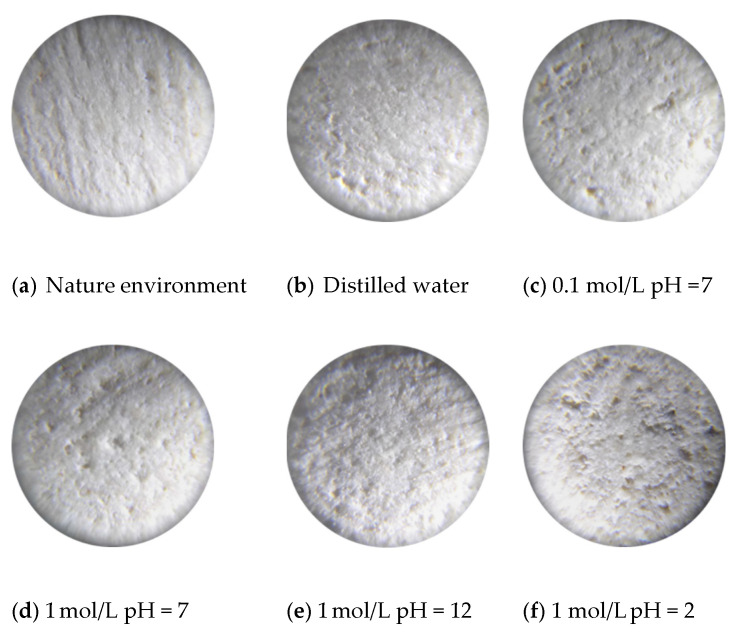
30-fold enlargement of fissure surface under different chemical corrosion.

**Figure 6 sensors-21-05641-f006:**
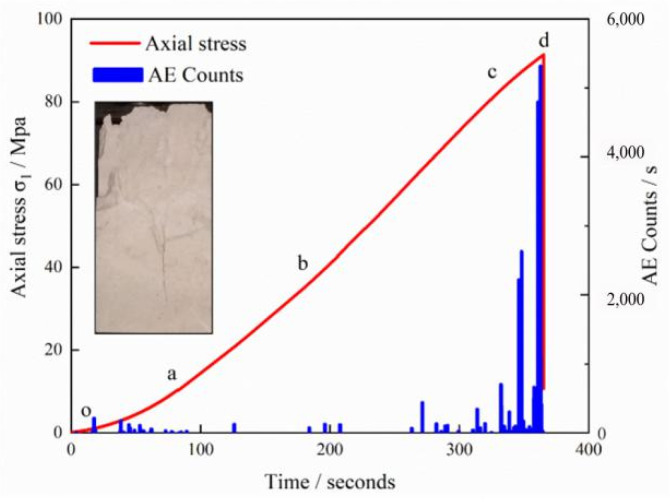
Relation between AE counts, axial stress and time of intact limestone sample.

**Figure 7 sensors-21-05641-f007:**
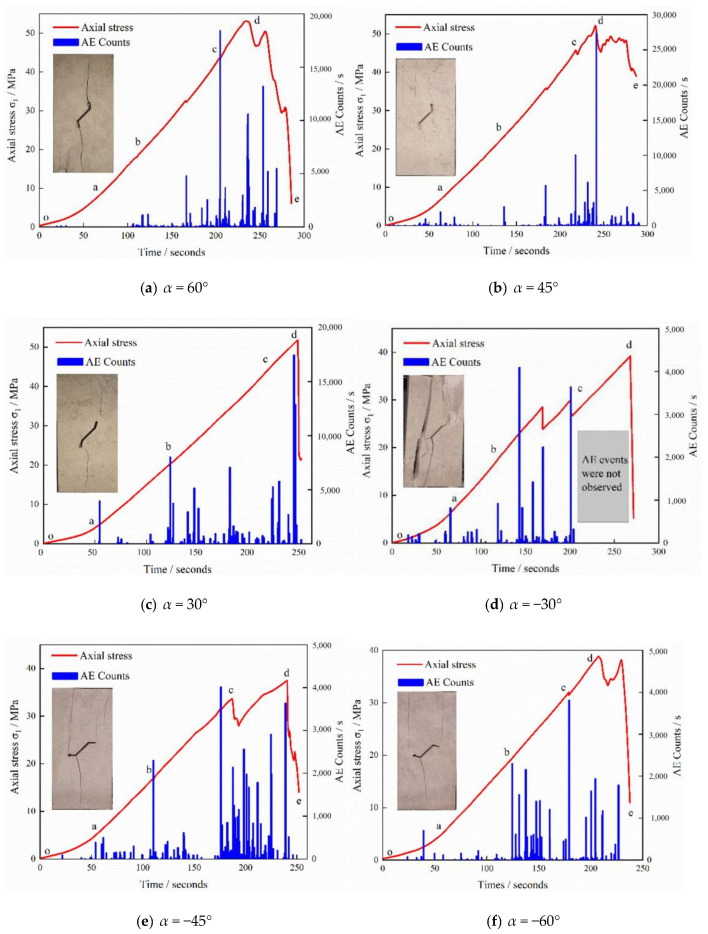
Relation between AE counts, axial stress and time of limestone sample containing single kinked fissure.

**Figure 8 sensors-21-05641-f008:**
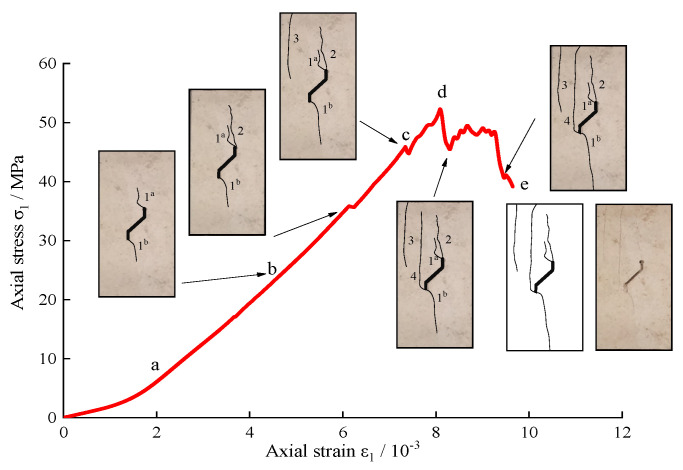
Axial stress-strain curve and the cracking process of limestone containing single kinked fissure. (*α* = 45°).

**Figure 9 sensors-21-05641-f009:**
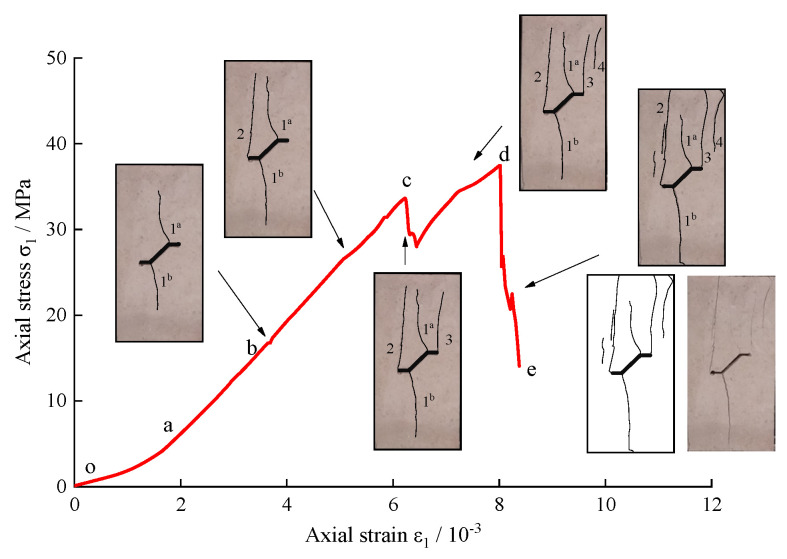
Axial stress-strain curve and the cracking process of limestone containing single kinked fissure. (*α* = −45°).

**Figure 10 sensors-21-05641-f010:**
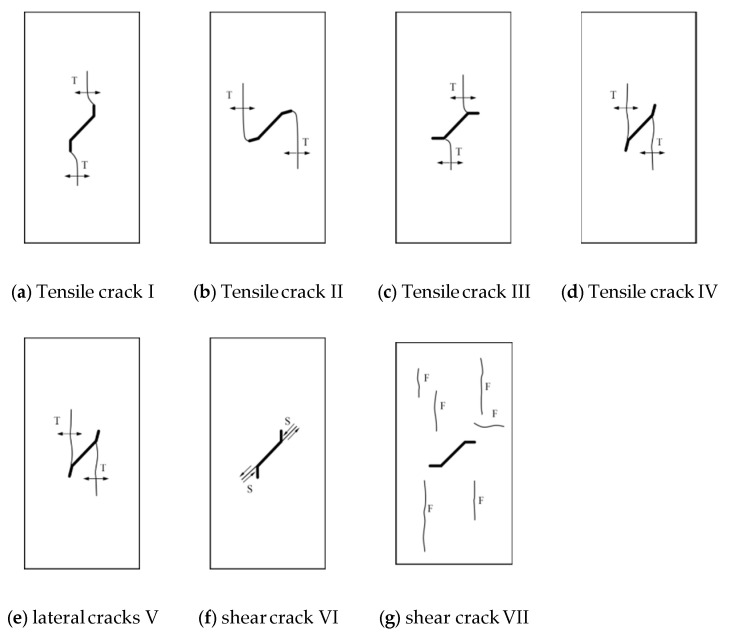
Various crack types initiated from the pre-existing kinked fissures identified in the present study.

**Figure 11 sensors-21-05641-f011:**
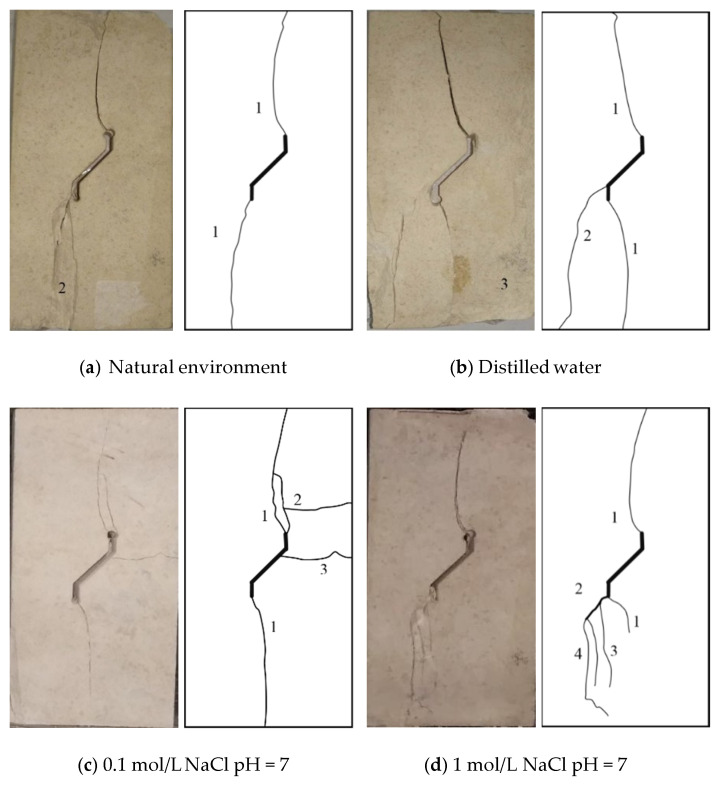
Effect of chemical corrosion on final failure modes of specimens containing single kinked fissure under uniaxial compression (*α =* 45°).

**Table 1 sensors-21-05641-t001:** Basic physical properties of rocks and specific mineral contents.

Natural Density (g/cm^−3^)	Natural Moisture Content	Mineral Composition				Uniaxial Compressive Strength/Mpa
		Calcite	Biochip	Quartz	Matal Mineral	Compressive Strength/Mpa
2.34	1.7%	68%	30%	1%	1%	89.8

**Table 2 sensors-21-05641-t002:** Chemical Solution Configuration.

Ingredient	Concentration	pH
NaCl	0.1 1	2 7 12
Distilled water	null	7

**Table 3 sensors-21-05641-t003:** Peak Stress of limestone containing single kinked fissure after different chemical corrosion under uniaxial compression.

Chemical Solution		*α* = +60°	*α* = +45°	*α* = +30°	*α* = −30°	*α* = −45°	*α* = −60°
Natural environment	*σ*	54.11	53.25	51.2	40.69	39.11	38.93
Distilled water	*σ^t^*	51.63	50.47	48.8	39.07	37.45	36.65
	*ρ*	4.58	5.23	4.68	4	4.23	5.86
0.1 mol/L NaCl pH = 2	*σ^t^*	43.16	42.61	42.16	30.8	29.92	30.05
	*ρ*	20.23	19.98	17.66	24.31	23.49	22.81
0.1 mol/L NaCl pH = 7	*σ^t^*	50.05	49.65	48.88	38.72	36.16	35.85
	*ρ*	7.49	6.76	4.53	4.85	7.53	7.91
0.1 mol/L NaCl pH = 12	*σ^t^*	48.65	46.37	45.00	36.65	33.29	33.49
	*ρ*	10.09	12.92	12.11	9.93	14.87	13.97
0.1 mol/L NaCl pH = 2	*σ^t^*	41.62	40.47	40.69	29.88	29.73	29.73
	*ρ*	23.07	24	20.52	26.57	23.97	23.63
0.1 mol/L NaCl pH = 7	*σ^t^*	49.78	48.76	48.07	38.07	36.05	35.49
	*ρ*	8.00	8.43	6.11	6.45	7.82	8.84
0.1 mol/L NaCl pH = 12	*σ^t^*	46.81	45.57	44.52	34.07	32.93	32.65
	*ρ*	13.48	14.42	13.05	16.28	15.79	16.13

Note: $\rho =\frac{\sigma -\sigma^{\prime }\;}{\sigma \;}\times 100\%$.

**Table 4 sensors-21-05641-t004:** Initiated crack types of limestone samples containing single kinked fissure with different chemical corrosion.

Crack Patterns	Chemical Environment					
	Natural Environment	Distilled Water	0.1 mol/L NaCl pH = 7	1 mol/L NaCl pH = 7	1 mol/L NaCl pH = 12	1 mol/L NaCl pH = 2
Pattern I	√1	√1	√1	√1	√1	√1
Pattern II	√					
Pattern III						
Pattern V						√
Pattern VI			√	√	√	
Pattern VII		√			√	√
Pattern VIII	√	√	√	√	√	√

## Data Availability

The data presented in this study are available on request from the corresponding author.

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
