# Peer review of "The Effect of Chemical Corrosion on Mechanics and Failure Behaviour of Limestone Containing a Single Kinked Fissure"

_sensors, 2021, doi:10.3390/s21165641_

Round 1

Reviewer 1 Report

This study gives the influence of chemical corrosion and geometric parameters on the macroscopic damage characteristics of brittle limestone containing a kinked fissure under uniaxial compression. It is interesting for scientific research about the considerations on both corrosion and mechanics. Before acceptance, there are some comments as below:

  1. The experimental procedure seems to be that it is first to do the chemical solutions and then to do the compression tests, but not to do both of these two tests at the same time. It is right? If it is right, the research method can reveal the joint influence on crack? Please give an address.
  2. This research involves the different α, can the authors consider the change of “45º”. Some research papers can be seen as “Frontiers of Structural and Civil Engineering. 13 (2019) 288-293” and International Journal of Impact Engineering 106 (2017) 217-222”. Both these two angles have an effect on the strength.
  3. The Theoretical calculation of stress intensity factor of kinked crack only considers the chemical corrosion that induces the change of the length of the branch fissure and the material fracture stress. If some more insight influences and physical parameters can be given for addressing the chemical corrosion? Please think about it.

Reviewer 2 Report

  1. In this paper, a uniaxial compression test was performed. It is necessary to explain why the uniaxial compression test was carried out.

  1. It is necessary to explain with a detailed picture of sample in Figure 3.

  1. Line 133-134: Was the high-speed camera taking video? Or were images taken? A detailed description of monitoring is required, such as the type of camera and data acquisition frequency.

  1. Figure 7 shows the main measurement results of this paper. Each case needs to be described in detail.

  1. In Figures 7a, 7d, 7e, and 7f, the AE results are highest around point c. What are the reasons for this? Can you explain the results of axial stress and AE in relation to each other?

  1. Section 4

Why is the theoretical formula presented in Section 4? A conclusion should be drawn by linking the previous test results with the theoretical formula.

  1. Conclusion 1

Line 326-328: “Moreover, the chemical environment has a minor effect on the compressive strength of rock based on chemical tests, and this effect is shown in the order of natural environment > distilled water > alkaline environment > acidic environment.”

A clear rationale for making this conclusion should be described in detail.

  1. Conclusion 4

Line 343-344: “Based on the theory of fracture mechanics, the stress intensity factor of kinked fissures under chemical corrosion is obtained.”

In this paper, the theoretical formula was introduced, but it was not connected with the results of the test. A conclusion should be drawn by clearly defining the relationship between the theoretical formula and the experimental results.

Acoustic emission (AE) technique and high-speed camera were applied in this paper to measure cracks. Recently, techniques for measuring cracks using thermal infrared cameras or using deep learning or machine learning techniques are being used. A discussion of these latest technologies is necessary in the introduction, and the following papers contain the latest technologies.

- Seo et al. (2021) Crack detection in pillars using infrared thermographic imaging, Geotechnical Testing Journal.

  1. Lines 286, 312, 315, Equation 9 and 13:

It is not appropriate to construct a sentence with ‘we’ as the subject.

Round 2

Reviewer 2 Report

The authors addressed appropriate responses to the reviewer's comments.